supramolecular chemistry/computational chemistry/nanotechnology

click reaction, hydrogel, azide, acetylene, HOMO, LUMO

**Author for correspondence:**
Abu Bin Imran
e-mail: abimran@chem.buet.ac.bd

This article has been edited by the Royal Society of Chemistry, including the commissioning, peer review process and editorial aspects up to the point of acceptance.

# Facile fabrication of polymer network using click chemistry and their computational study

## Md. Kausar Ahmed, Ajoy Kumer and Abu Bin Imran

Department of Chemistry, Faculty of Engineering, Bangladesh University of Engineering and Technology, Dhaka 1000, Bangladesh

ABI, 0000-0001-8920-8921

Click reaction is a very fast, high yield with no by-product, biocompatible, tolerant to surrounded medium, and very specific cycloaddition reaction between azides and alkynes to form triazole. They are widely being employed in the synthesis of various polymeric materials. Here, the design, fabrication and characterization of hydrogel prepared using click reaction have been reported. At first, telechelic acetylene precursor for click reaction is prepared from diisocyanatohexane and propargyl alcohol in the presence of triethylamine. The azide derivatives of poly(hydroxyethylmethacrylate), i.e. poly(HEMA), are successfully prepared following two different routes. In route 1, esterification of bromopropionic acid is performed with HEMA monomer using *N,N′*-dicyclohexylcarbodiimide/4-dimethylaminopyridine (DCC/DMAP) as a catalyst followed by replacing bromide by azide moiety. Free radical polymerization of the fabricated monomer is then performed under $N_2$ atmosphere using azobisisobutyronitrile (AIBN) as an initiator. In route 2, polymerization of HEMA has been carried out first, then modification of the polymer with azide group via successive steps to obtain azide derivative polymer for click reaction. The hydrogel is prepared by a very fast, highly specific, and simple click reaction between azide derivative polymer and telechelic acetylene precursor using copper as a catalyst. The structures of derivatives of azide-functionalized HEMA, acetylene precursors and hydrogels are confirmed by FTIR and $^1$H-NMR spectroscopy. The optimized structure of each precursor is determined, and their chemical and thermodynamic parameters are computationally studied in detail.

## 1. Introduction

A hydrogel is three-dimensional polymer networks that absorb a significant volume of water without dissolving in any solvents [1].

## ROYAL SOCIETY OF CHEMISTRY

It maintains its three-dimensional integrity by physical or chemical cross-linking of the polymers. Physical networks of hydrogel develop mainly by H-bonding, hydrophobic interactions, stereo- and supra-molecular complex formation, ionic interactions and so on [2–4]. On the other hand, chemical cross-linking permanently connects the polymers through irradiation of rays, small cross-linker molecules, polymer–polymer cross-linking, condensation reaction, ionization, etc. [5–7]. Hydrogels are widely being used in biomedical areas, applications in drug delivery systems, tissue engineering, smart sensor and so on [8,9]. Despite their extensive potential applications, the low mechanical resistance and poor stimuli sensitivity resulting from the micro-level network inhomogeneities developed during uncontrolled nature cross-linking reaction restricts their widespread uses. In the case of typical cross-linking via free-radical polymerization, the extent of heterogeneity of polymer network is influenced by the concentration of cross-linker, or monomer or initiator, and reaction temperature. The hydrogel with inhomogeneous polymer networks cannot dissipate energy under applied stress to exhibit improved mechanical performance. In order to achieve energy dissipation under applied stress, which is a prerequisite criterion to obtain mechanically strong and fast responsive hydrogels, a number of attempts have been made, for example, nanocomposite gel using disc-shaped nanoclay [10], a double-network gel having two individual networks [11], polyrotaxane gel that contains sliding cross-linkers [12,13], and so on. Sakai *et al.* developed a *tetra*-PEG gel by combining two symmetrical tetrahedron-like macromonomers to obtain homogeneous polymer networks having high mechanical strength comparable to that of native articular cartilage [14]. They also reported cross-linked hydrophilic poly(ethylene glycol) (linear-PEG) and hydrophobic poly(dimethylsiloxane) (linear-PDMS) prepolymers by 'macromolecular' PEG cross-linker (*tetra*-PEG) to prepare homogeneous hydrogel network [15]. Malkoch *et al.* investigated PEG-based hydrogel by click chemistry results in an ideal network structure leading to improve properties compared to traditional photochemically cross-linked PEG [16]. They first prepared acetylene-functionalized PEG polymer by esterification of hydroxy-terminated PEGs having varying molecular weights with 4-pentynoic anhydride in the presence of 4-dimethylaminopyridine. Another precursor, i.e. tetraazide was produced by activation of tetrahydroxy PEG derivative with mesyl chloride followed by nucleophilic substitution of sodium azide. The tetrahydroxy PEG derivative was synthesized by quantitative esterification of tetraethylene glycol with the anhydride of isopropylidene-2,2-bis(methoxy)propionic acid. Finally, the click reaction of tetraazide with 2.0 equivalents of PEG diacetylene at room temperature in the presence of copper sulfate and sodium ascorbate produced a well-defined hydrogel network.

Click reaction, a cycloaddition of azides and alkynes leads to form triazole. It is a high yielding reaction and produces a single by-product which can be eliminated without using chromatography or recrystallization process. The click reaction is environmentally stable, tolerant of various additives, and it exhibits consistent reactivity in the presence of a wide variety of functional groups under physiological surroundings [17,18]. The reaction rate of copper-catalysed azide-alkyne cycloaddition is $10^7$ to $10^8$ times higher relative to uncatalysed cycloaddition [19–21]. The cycloaddition can also be performed by ruthenium-catalysed process [22]. Hydrogels synthesized by click reaction form homogeneous network and overcome the inherent limitations of conventional hydrogels [23,24]. It is an attractive way to synthesize hydrogels, which could blend the best functionality with improved chemical control and diversity. Cross-linking can spread symmetrically in the network with uniform network distribution and can withstand applied pressure. However, the preparation of *tetra*-arm shaped precursors for click reaction mentioned by the group is costly, time-consuming and difficult.

Here, we choose 2-hydroxyethyl methacrylate (HEMA) as a monomer, which is exceedingly functional, environmentally benign and biocompatible for its potential applications in biomedical areas [25,26]. The functional groups of HEMA monomer or polymer can easily be modified into azide precursor, which gives more freedom to tailor additional functionality after the hydrogel formation. The bifunctional instead of *tetra*-arm or *tetra*-functional acetylene precursor has easily been synthesized, and click reaction between two precursors are employed to form the hydrogel network. The thermodynamics, chemical stability, chemical reactivity and toxicity data of the precursors and click hydrogel have been analysed using quantum computation tools that are imperative for further development.

## 2. Experimental

**Synthesis of di(prop-2-yn-1-yl) hexane-1,6-diyldicarbamate (1)**: A 100 ml round bottom flask was filled with 10 ml of dry THF with magnetic stirring, and then triethylamine (1.26 g, 12.45 mmol), propargyl

alcohol (1.54 g, 27.39 mmol) and 1,6-diisocyanatohexane (2.1 g, 12.45 mmol) were added to the reaction vessel. The reaction vessel was immersed in an ice bath and stirred for 24 h. It was then quenched with 2 M HCl and cleaned by ethyl acetate. Anhydrous $MgSO_4$ powder was used to dry the organic part by magnetic stirring for 30 min. Then it was filtered, and the solvent was removed at reduced pressure to obtain oily residue of compound (**1**) with a 79% yield.

IR (KBr, $v$/cm$^{-1}$): 1687 (C=O stretching), 2945 (C–H stretching of $CH_3$ group), 1146 (C–O stretching), 2131 (C≡C stretching) and 3310 (N–H stretching).

$^1$H-NMR (CDCl$_3$, $\delta$, ppm): 4.69 (for –C≡CH proton), 3.18–3.23 (–$CH_2$-N– protons), 7.28 (N–H proton), 4.83 (for –$OCH_2$ protons) and 1.27–1.62 (for –$CH_2CH_3$ protons).

**Synthesis of 2-((2-bromopropanoyl)oxy)ethyl methacrylate (2)**: A 50 ml round bottom flask was filled with 2-bromopropinoic acid (200 mg, 1.31 mmol), DCC (276 mg, 1.34 mmol) and dichloromethane (4 ml). The reaction mixture was cooled in an ice bath. HEMA (170 mg, 1.31 mmol), DMAP (0.02 g) and dichloromethane (4 ml) solution were added dropwise for 30 min under vigorous stirring. The reaction mixture was then permitted to swirl for 1 h at 0°C and then at room temperature for 18 h. The insoluble compound was separated by filtration, and the filtrate was extracted with dichloromethane followed by washing multiple times with saturated NaHCO$_3$ aqueous solution and pH 1 aqueous solution in a separatory funnel. The organic part was dried by adding anhydrous MgSO$_4$ powder. After filtration, the solvent was removed at reduced pressure to obtain oily residue of compound (**2**) with a 62% yield.

IR (KBr, $v$/cm$^{-1}$): 1743 (C=O stretching), 2930 (C–H stretching of $CH_3$ group), 1627 (C=C stretching) and 1157 (C–O stretching).

$^1$H-NMR (CDCl$_3$, $\delta$, ppm): 4.35 (for –C–H proton), 5.53 and 6.06 (for HC=CH protons), 3.78–4.22 (for –$OCH_2$ protons) and 1.74–1.87 (for –$CH_2CH_3$ protons).

**Synthesis of 2-((2-azidopropanoyl)oxy)ethyl methacrylate (3)**: The compound (**2**) (340 mg, 1.28 mmol), NaN$_3$ (160 mg, 2.46 mmol) and 8 ml DMF were added into a 50 ml round bottom flask. The reaction mixture was then allowed to simmer at room temperature for 12 h. After completion of the reaction, dichloromethane was added to the mixture and extracted by deionized water. The sample was dried and collected following the same procedure mentioned earlier for compound (**2**), and the total yield is 58%.

IR (KBr, $v$/cm$^{-1}$): 1746 (C=O stretching), 2933 (C–H stretching of $CH_3$ group), 1628 (C=C stretching) and 1165 (C–O stretching).

**Synthesis of poly[2-((2-azidopropanoyl)oxy)ethyl 2-methylbutanoate] (4)**: DMF (7 ml) was taken into a 50 ml round bottom flask and degassed for 20 min. The synthesized compound (**3**) (150 mg, 0.66 mmol) and initiator AIBN (1%) were carefully added to the reaction flask. The reaction mixture was permitted to swirl under the nitrogen atmosphere at 60°C for 6 h. The crude product was removed using dichloromethane and water. The solvent was removed by rotary evaporator; a pure and sticky type compound (**4**) was obtained with a 53% yield.

IR (KBr, $v$/cm$^{-1}$): 1758 (C=O stretching), 2932 (C–H stretching of $CH_3$ group) and 1167 (C–O stretching).

$^1$H-NMR (CDCl$_3$, $\delta$, ppm): 2.97 and 2.89 (for –C–H proton), 3.73–4.78 (for –$OCH_2$ protons) and 1.31–1.95 (for –$CH_2CH_3$ protons).

**Synthesis of hydrogel (5) by click reaction**: The compound (**4**) (110 mg), compound (**1**) (7 mg, 0.018 mmol) and 120 mg ethanol were taken into a 50 ml round bottom flask. Sodium ascorbate (2 mg) and 2 mg of copper sulfate were added into 120 mg of deionized water. The hydrogel (**5**) was formed within 5 min. The hydrogel was then vigorously washed with 5% EDTA solution to remove the trapped copper from the networks and then washed with plenty of deionized water for 7 days to obtain pure hydrogel.

IR (KBr, $v$/cm$^{-1}$): 1688 (C=O stretching), 2942 (C–H stretching of $CH_3$ group), 1146 (C–O stretching) and 3313 (N–H stretching).

**Synthesis of poly[2-hydroxyethyl 2-methylbutanoate] (6)**: DMF (10 ml) was taken into a 50 ml round bottom flask and degassed for 20 min. The HEMA (170 mg, 1.307 mmol) and initiator AIBN (2 mg) were added into the reaction flask, and the solution was swirled under the nitrogen atmosphere at 75°C for 24 h. The crude polyHEMA was extracted using hexane and water, and dried by a rotary evaporator. The pure and sticky compound (**6**) was collected with a 55% yield.

IR (KBr, $v$/cm$^{-1}$): 1745 (C=O stretching), 2933 (C–H stretching of $CH_3$ group), 1154 (C–O stretching) and 3338 (O–H stretching).

**Synthesis of poly[2-((2-bromopropanoyl)oxy)ethyl 2-methylbutanoate] (7)**: The compound (**7**) was prepared following the same synthetic procedure described for compound (**2**). 2-bromopropinoic acid

(597 mg, 3.902 mmol), compound (**6**) (507 mg, 3.902 mmol), DCC (505 mg, 3.99 mmol), DMAP (0.06 g) and dichloromethane (20 ml) were used to synthesize compound (**7**) with 60% yield.

IR (KBr, $v$/cm$^{-1}$): 1744 (C=O stretching), 2932 (C−H stretching of CH$_3$ group) and 1158 (C−O stretching).

**Synthesis of poly[2-((2-azidopropanoyl)oxy)ethyl 2-methylbutanoate] (8)**: The compound (**8**) was prepared following the same synthetic procedure described for compound (**3**). Here, compound (**7**) (55 mg), NaN$_3$ (50 mg) and DMF 12 ml were used to obtain compound (**8**) with a 63% yield.

IR (KBr, $v$/cm$^{-1}$): 1744 (C=O stretching), 2932 (C−H stretching of CH$_3$ group) and 1155 (C−O stretching).

$^1$H-NMR (CDCl$_3$, $\delta$, ppm): 2.97 and 3.83 (for −C−H proton), 3.90–4.68 (for −OCH$_2$ protons) and 1.07–1.95 (for −CH$_2$CH$_3$ protons).

**Synthesis of hydrogel (9) by click reaction**: The click reaction between compound (**8**) (80 mg) and compound (**1**) (5 mg) in the presence of sodium ascorbate (2 mg), copper sulfate (2 mg), ethanol (100 mg) and deionized water (100 mg) produced a hydrogel (**9**).

IR (KBr, $v$/cm$^{-1}$): 1694 (C=O stretching), 2944 (C−H stretching of CH$_3$ group), 1145 (C−O stretching) and 3312 (N−H stretching).

# 3. Materials and characterizations

All chemicals and reagents in this work were analytical grade and used without further purifications. In order to confirm the structure, the Fourier transform infrared (FT-IR) spectrophotometer (FTIR-8400, Shimadzu, Japan) was used in the 4000–500 cm$^{-1}$ region. At 60°C, the samples were oven-dried and mixed grinding samples with pure KBr (Sigma-Aldrich, Germany) crystals to perform FT-IR spectra of the solid and powdered samples. The powder mixture was then compressed manually to form a pellet and placed in the sample chamber for measurement. By analysing the nuclear magnetic resonance ($^1$H-NMR) spectra, the structures of the compounds synthesized in various steps were confirmed. The 400 MHz Bruker BPX-400, $^1$H-NMR spectrometer was used to analyse chemical shifts ($\delta$) in ppm and the coupling constant (j) in Hz using CDCl$_3$ as a solvent and tetramethylsilane (TMS) as an internal standard. The chemical shifts of samples were recorded compared to the protons peak of TMS. Simultaneous thermogravimetry–differential scanning calorimetry (STA/TG-DSC) of STA 449 F3 Jupiter®, NETZSCH-Gerätebau GmbH, Germany were used to perform thermal studies. Approximately 5 mg of dried and powdered sample was taken in a platinum sample pan and analysed from room temperature to 900°C under a nitrogen atmosphere at a heating rate of 10°C min$^{-1}$. The nitrogen gas was purged at a flow rate of 20 ml min$^{-1}$. For optimization of the structure, calculation of vibrational frequency and molecular orbital for molecules, the most common functional DFT was employed for quantum calculations [27]. The VAMP code of Materials Studio version 8.0 was employed for optimization and calculation based on DFT [28,29]. After optimization, the HOMO, LUMO frontier molecular orbitals and their magnitudes were analysed to obtain electrostatic potential map, electron density and thermodynamics, and so on. After analysis, the compounds were converted into a SMILES file for toxicity evaluation which was performed by amdetSAR online database system.

# 4. Results and discussion

The telechelic acetylene derivative (**1**) was prepared with a 79% yield by the reaction of 1,6-diisocyanatohexane with propargyl alcohol in the presence of triethylamine at room temperature for 24 h in dry THF (figure 1). The characteristic absorption bands of compound (**1**) are observed at 2945, 1687, 1146, 2131 and 3310 cm$^{-1}$ for the $v$(C−H), $v$(C=O), $v$(C−O), $v$(C≡C) and $v$(N−H), respectively. The $^1$H-NMR spectrum of compound (**1**) displays a peak at $\delta$ 4.69 ppm due to the successful incorporation of C≡CH proton. The other characteristic peaks at $\delta$ 3.18–3.23 ppm for (−CH$_2$−N−) protons, peak at $\delta$ 4.83 ppm for alkoxy protons, peaks at $\delta$ 1.27–1.62 ppm for alkyl protons and peak at $\delta$ 7.28 ppm for (N−H) proton also appear.

The azide-functionalized polyHEMA (**4/8**) was synthesized via two different routes that differ only in the polymerization initiation time of HEMA (scheme 1). In the first route, polymerization of HEMA was performed after esterification and insertion of azide functionalization, whereas in the second route, polymerization of HEMA was carried out first prior to the azide functionalization. The 2-((2-bromopropanoyl)oxy)ethyl methacrylate (**2**) was prepared by reaction between 2-bromopropinoic acid and HEMA in dichloromethane solvent for 18 h at room temperature using

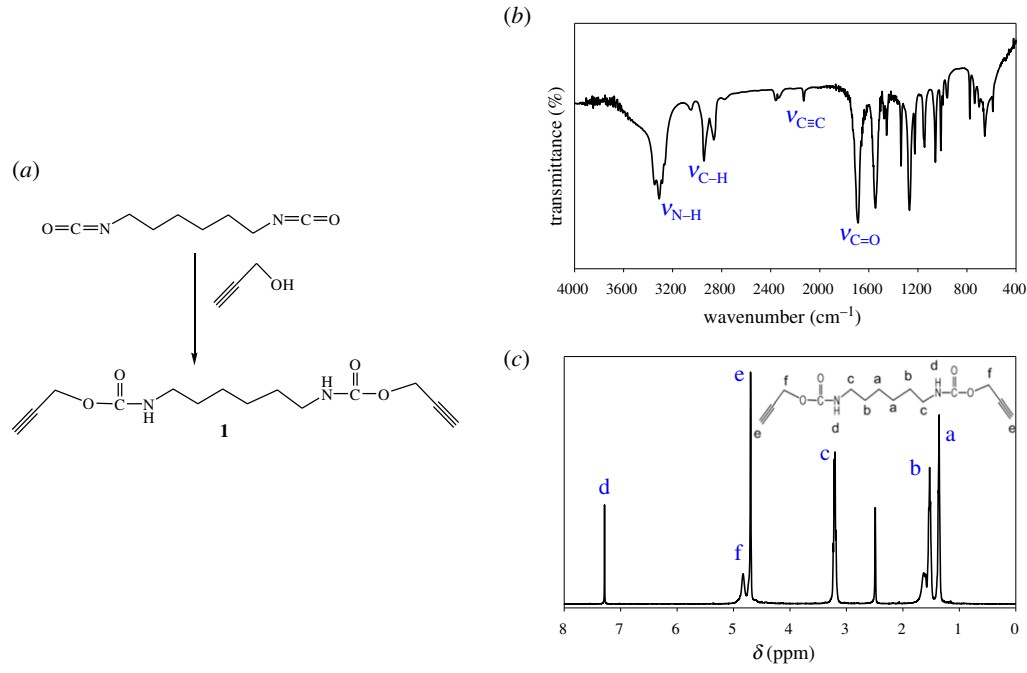

**Figure 1.** (a) Synthesis scheme, (b) FT-IR spectra and (c) ¹H-NMR spectra of diacetylene derivative (1).

dicyclohexylcarbodiimide/4-dimethylaminopyridine (DCC/DMAP) as a catalyst. The characteristic absorption bands of compound (**2**) are observed at 2930, 1743, 1157 and 1627 cm$^{-1}$ for $v$(C−H), $v$(C=O), $v$(C−O) and $v$(C=C), respectively (figure 2*a*). From ¹H-NMR spectra in figure 3, the peak at $\delta$ 4.35 ppm for the (C−H) protons, the peaks at $\delta$ 5.53 and $\delta$ 6.06 ppm for the vinyl protons, the peaks at $\delta$ 3.78–4.22 ppm for alkoxy protons, and the peaks at $\delta$ 1.74–1.87 ppm for alkyl protons, respectively are distinctly observed. The insertion of azide group into compound (**2**) was simply achieved by reacting with NaN$_3$ in dimethylformamide (DMF) solvent at room temperature for 12 h. The dichloromethane was found not a suitable solvent for this reaction, as at high temperatures, the azide coupling reactions stopped there. The characteristics absorption bands at 2933, 1746, 1165 and 1628 cm$^{-1}$ for the $v$(C−H), $v$(C=O), $v$(C−O) and $v$(C=C), respectively are observed in the FT-IR spectrum that confirm the formation of compound (**3**). The low intensity absorption peak for azide groups of compounds (**3**) and (**4/8**) are observed only after magnification of the curves in the region of 2120 cm$^{-1}$. The free-radical polymerization of compound (**3**) was then accomplished in DMF solvent using $\alpha,\alpha'$-azoisobutyronitrile (AIBN) as an initiator at 60°C for 6 h under nitrogen atmosphere to obtain poly[2-((2-azidopropanoyl)oxy)ethyl 2-methylbutanoate] (**4**). Under the same conditions, increasing reaction time leads to produce an insoluble polymer. The reaction was not progressed at room temperature even for 24 h reaction time (electronic supplementary material, table S1). The use of an initiator 2,2'-azobis(2-methylpropionamidine)dihydrochloride (V-50) instead of AIBN at 60°C was also not successful in producing the compound (**4/8**). There is no absorption band observed for C=C bond in the FT-IR spectrum for compound (**4**) that indicates the successful polymerization of HEMA (figure 2*a*). Other characteristic peaks at 2932, 1758 and 1167 cm$^{-1}$ for the $v$(C−H), $v$(C=O) and $v$(C−O), respectively, are observed. The peaks of (C−H) proton at $\delta$ 2.97 and $\delta$ 2.89 ppm, alkoxy protons at $\delta$ (3.73–4.78) ppm and alkyl protons at $\delta$ (1.31–1.95) ppm are observed in the ¹H-NMR spectrum. As expected, the peaks for vinyl protons are disappeared after polymerization that confirms the successful polymer formation. The click reaction between compound (**4**) and compound (**1**) using CuSO$_4$.5H$_2$O as a catalyst, sodium ascorbate as reducing agent and H$_2$O/EtOH as solvent produce hydrogel (**5**). CuSO$_4$.5H$_2$O and sodium ascorbate are found to generate *in situ* an active Cu$^{+1}$ in the reaction mixture by reducing Cu(II) salts. The thermodynamic instability of Cu$^{+1}$ is essential for its addition to a combination of reactions. The gel formation was completed within 5 min, and copper was removed from the gel by washing with 5% ethylenediaminetetraacetic acid (EDTA) aqueous solution. Hydrogel (**5**) formation was initially confirmed by checking the swelling ability of the materials in different solvents having varying dielectric constant. The characteristic absorption bands in FT-IR spectra of hydrogel (**5**) are observed at 2942, 1688, 1146 and 3313 cm$^{-1}$ for the $v$(C–H), $v$(C=O), $v$(C–O) and $v$(N–H), respectively. In the second approach, the polymerization of HEMA was

**Scheme 1.** Synthetic scheme to prepare the hydrogel (**5/9**) using click reaction.

performed first in DMF solvent using AIBN as an initiator at 75°C for 24 h (scheme 1). The use of V-50 as an initiator at 60°C to 75°C for 6–18 h was unable to produce polyHEMA (**6**) (electronic supplementary material, table S2). There is no absorption band of C=C bond observed for compound (**6**) in FT-IR spectrum (figure 2*b*) that indicates the complete polymerization of the monomer. Other characteristics bands of compound (**6**) are also noted at 2933, 1745 and 1154 cm$^{-1}$ for the $v$(C–H), $v$(C=O) and $v$(C–O), respectively. The rest of the steps, i.e. esterification of poly-HEMA (**7**), azide insertion (**8**) and hydrogel (**9**) synthesis, are following the same synthetic procedure as mentioned in route 1. The successful formation of compound (**7**), (**8**) and hydrogel (**9**) are likewise confirmed from the FT-IR and $^1$H-NMR spectra.

The quantum calculation using VAMP code of Materials Studio was successfully exploited to predict the geometry, electronic structure and reactivity of the synthesized compounds [30–35]. From the frontier molecular orbitals, the HOMO, LUMO and electrostatic potential mapping are obtained to understand the possible electronic transition of the compounds and to find the electrophilic and nucleophilic interaction area in the molecule (figure 4). The value of HOMO for all molecules is found in the range from −10.324 to −6.005 eV, while for LUMO, it is ranging from −0.053 to −3.972 eV (table 1). The small energy difference of HOMO–LUMO ($\Delta$E) is considered as high chemical stability of compounds,

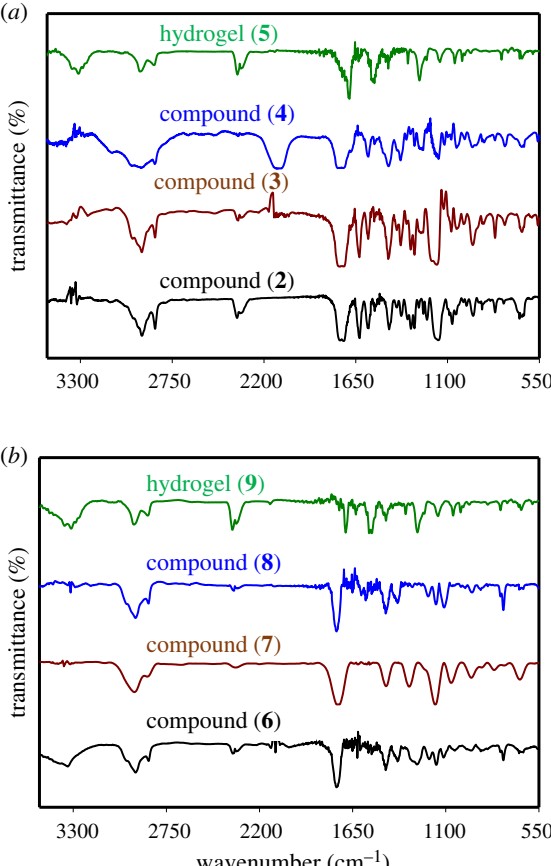

**Figure 2.** FT-IR spectra of (a) compound (**2**), compound (**3**), compound (**4**) and hydrogel (**5**), (b) compound (**6**), compound (**7**), compound (**8**) and hydrogel (**9**).

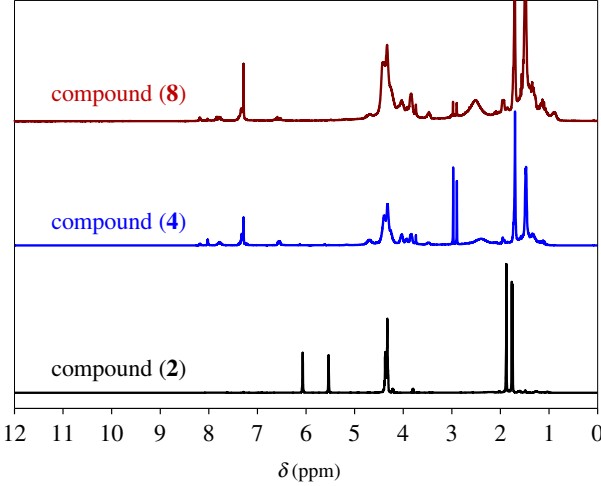

**Figure 3.** $^{1}$H-NMR spectra of compound (**2**), compound (**4**) and compound (**8**).

which leads to their biomedical applications [36–38]. The ΔE values for monomers are in the range of 8.0 to 10.0 eV while it is changed for polymers (**4/8**). A similar trend is found for compound (**6**) and compound (**7**). The ionization potential, electron affinity, electrophilicity, chemical potential and electronegativity values also validate the specific chemical reactivity of the compounds (table 1). Figure 4 illustrates the HOMO and LUMO frontier orbitals, HOMO lies around the electronegative atom, oxygen and in alkyl chains, and LUMO belongs to the electropositive atom, particularly nitrogen atom, and also in alkyl chains, although the polymer skeleton or body is completely free from both of HOMO and LUMO parts. The hardness of these compounds is recorded in the range of

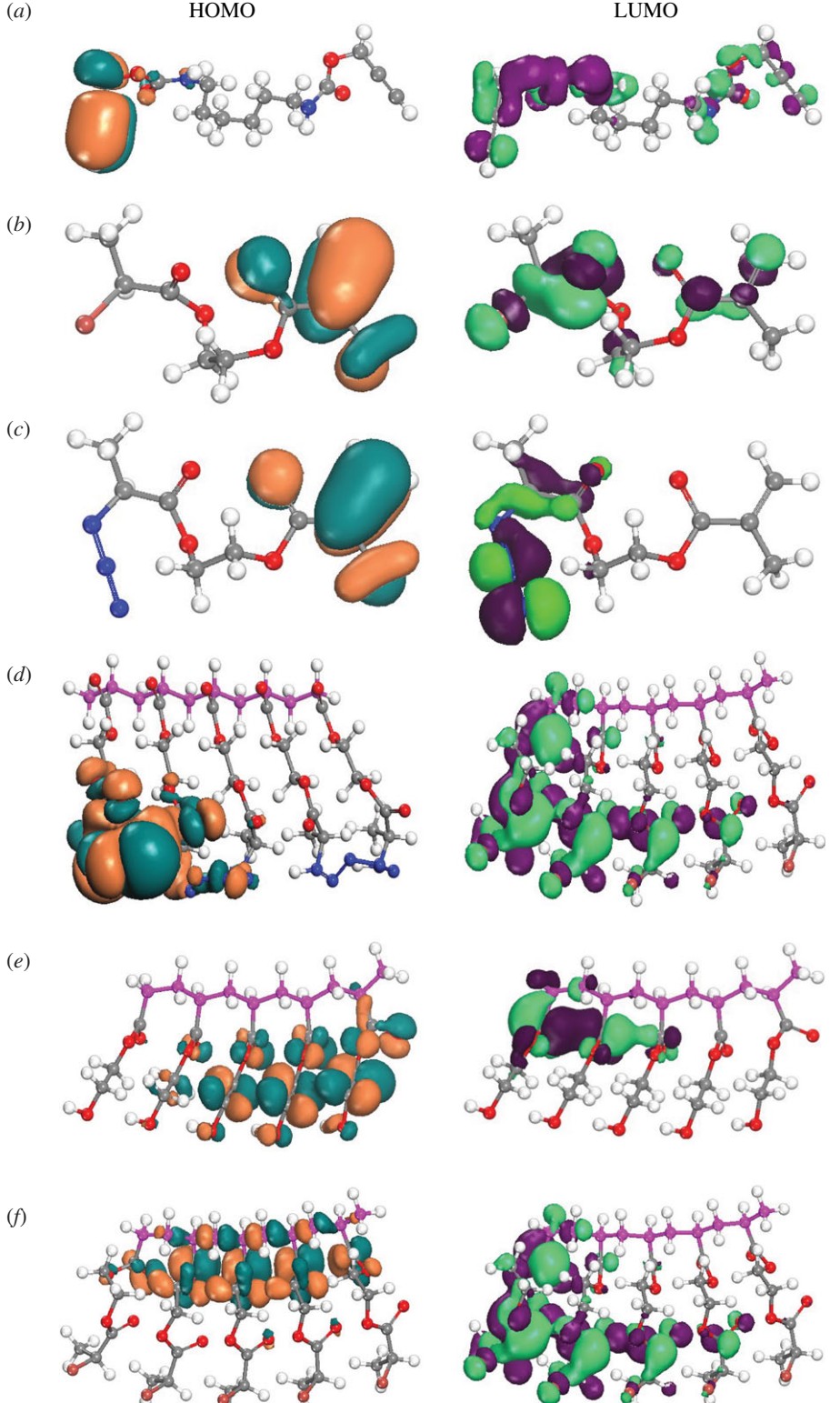

**Figure 4.** HOMO, LUMO orbital diagrams of (*a*) compound (**1**), (*b*) compound (**2**), (*c*) compound (**3**), (*d*) compound (**4/8**), (*e*) compound (**6**) and (*f*) compound (**7**). For HOMO, the red colour denotes the positive node, and the deep blue colour denotes the negative node of an orbital. For LUMO, the green colour denotes the positive node, and the violet colour denotes the negative node of an orbital.

5.027 to 3.979, except compound (**4/8**), which gives enormous and sturdily supportive information about high chemical reactivity of click reaction for the precursors. The compound (**6**) has minimum binding energy, i.e. the highest binding affinity towards ligands or small drug molecules. The compound (**4/8**)

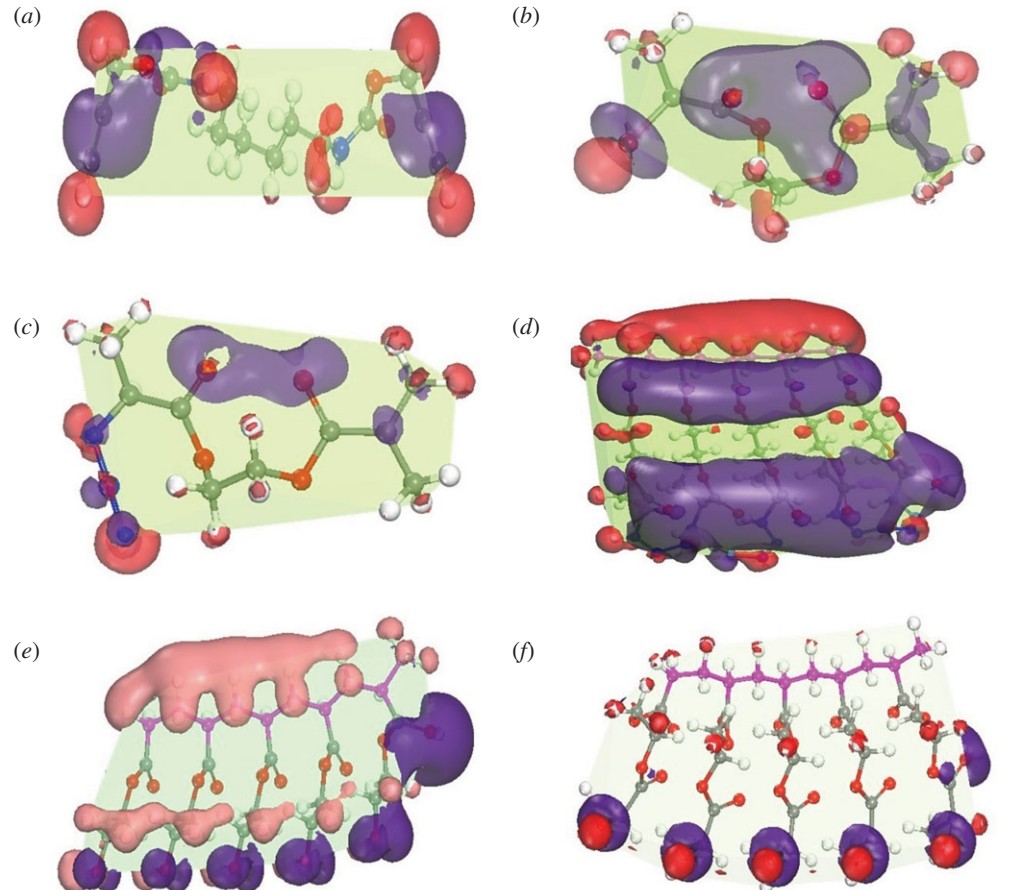

**Figure 5.** Electrostatic potential mapping of (*a*) compound (**1**), (*b*) compound (**2**), (*c*) compound (**3**), (*d*) compound (**4/8**), (*e*) compound (**6**) and (*f*) compound (**7**). The red colour denotes a positive charge region, the blue colour denotes a negative charge region, and the green colour denotes no charge region.

**Table 1.** HOMO, LUMO and chemical reactivities of compound (**1**), (**2**), (**3**), (**4**), (**6**), (**7**) and (**8**).

| chemical reactivity | 1 | 2 | 3 | 4/8 | 6 | 7 |
|---|---|---|---|---|---|---|
| HOMO (0), eV | −8.955 | −9.321 | −9.439 | −6.005 | −10.324 | −9.582 |
| LUMO, (0), eV | −0.053 | 0.734 | −1.481 | −3.972 | −1.240 | −0.481 |
| LUMO-HOMO gap, (ΔE), eV | 8.902 | 10.055 | 7.958 | 2.033 | 9.084 | 9.101 |
| ionization potential (I), eV | 8.955 | 9.321 | 9.439 | 6.005 | 10.324 | 9.582 |
| electron affinity (A), eV | 0.053 | −0.734 | 1.481 | 3.972 | 1.240 | 0.481 |
| hardness, ($\eta$), eV | 4.451 | 5.027 | 3.979 | 1.016 | 4.542 | 4.550 |
| softness, (S), eV$^{-1}$ | 0.225 | 0.199 | 0.251 | 0.984 | 0.220 | 0.220 |
| electrophilicity ($\omega$), eV | 2.278 | 1.833 | 3.746 | 12.244 | 3.680 | 2.781 |
| chemical potential, ($\mu$), eV | −4.504 | −4.293 | −5.460 | −4.988 | −5.782 | −5.031 |
| electronegativity, ($\chi$), eV | 4.504 | 4.293 | 5.460 | 4.988 | 5.782 | 5.031 |

and compound (**7**) have lower binding affinity due to their high binding energy. The free energy, entropy, dipole moment, binding energy, nuclear energy, electronic energy and heat of formation are calculated after the completion of optimization (electronic supplementary material, table S3). The compound (**7**) has a higher electrostatic potential energy difference than the others. The three-dimensional electrostatic potential mapping of all compounds shows the high surface area and charge region (positive and negative charge) (figure 5). The hydration energy of compound (**2**) is the lowest

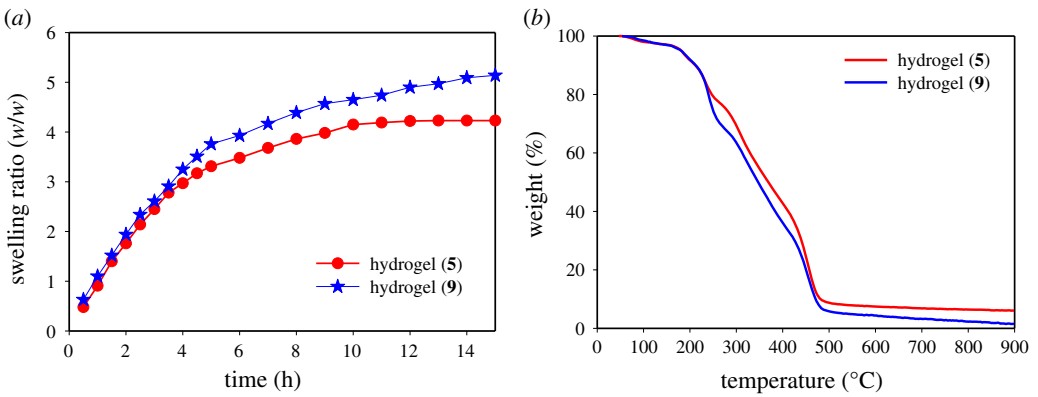

**Figure 6.** (a) Change in swelling ratios of hydrogel (**5**) and hydrogel (**9**) with time at room temperature, and (b) thermogravimetric analysis of hydrogel (**5**) and hydrogel (**9**).

(−1.71 kcal mol$^{-1}$), i.e. its solubility in water is high. But the compound (**4/8**) and compound (**7**) are poorly soluble in water and possess high hydration energy. The compound (**6**) has a negative partition function (logP) value of −1.14; therefore, it is hydrophilic, biocompatible and a suitable candidate for drug design. The rest of the compounds show a positive value of logP, which demonstrates their hydrophobic nature to be used unfavourably as a drug.

The online database admetSAR was used for the estimation of toxicological data, particularly aquatic and non-aquatic toxicity for all precursors [39]. The data (electronic supplementary material, table S6) show the biodegradable and non-biodegradable natures of the compounds so that they are less likely to harm the environment if they are mixed with any element of the environment even after their use. The compounds used in the study respond negatively to these human ether-a-go-go inhibition and carcinogenicity parameters.

The swelling ratios of hydrogels (**5/9**) at room temperature increase significantly with increasing time, and their equilibrium swelling ratio are found as 4.0 and 5.0, respectively (figure 6a). The swelling ratio of hydrogel (**9**) is slightly higher than hydrogel (**5**), as the polymerization of HEMA in the first step makes a more organized hydrophilic polymer that accommodates a large amount of water in the network. After water interaction with hydrophilic and hydrophobic sites, the osmotic driving force allows more water to be absorbed by the network. Finally, a level of swelling equilibrium is established by the balance between the retraction force and the infinite dilution force. Both hydrogels (**5/9**) reach their equilibrium swollen states within 15 h.

The initial weight loss of either hydrogels (**5/9**) is originated from the loss of bound water from the polymer networks. The decomposition temperature is in the range of 190–480°C for hydrogel (**5**) and 200–480°C for hydrogel (**9**), with negligible ash residue (figure 6b). The decomposition of polymer network for hydrogel (**5**) is at 480°C, which is slightly higher than hydrogel (**9**). The ash residue for hydrogel (**5**) is found 6%, which is marginally higher than hydrogel (**9**) with 1.5%.

# 5. Conclusion

The poly[2-((2-azidopropanoyl)oxy)ethyl 2-methylbutanoate] (**4**) polymer was synthesized by free radical polymerization of 2-((2-azidopropanoyl)oxy)ethyl methacrylate (**3**) monomer which is obtained by esterification reaction followed by insertion of azide group into HEMA monomer. The diacetylene precursor (**1**) is synthesized by a reaction of 1,6-diisocyanatohexane with propargyl alcohol in the presence of triethylamine at room temperature in a dry tetrahydrofuran solvent. The click reaction of compound (**4**) and compound (**1**) is performed in the presence of copper sulfate as catalyst, sodium ascorbate as reducing agent and ethanol/water as a solvent to prepare hydrogel (**5**). In the second route, poly HEMA (**6**) was prepared using AIBN as initiator at 75°C for 24 h under nitrogen atmosphere prior to esterification. After esterification, the bromo groups are replaced by azide to produce poly[2-((2-azidopropanoyl)oxy)ethyl 2-methylbutanoate] (**8**). The hydrogel (**9**) is prepared by the click reaction of compound (**8**) and compound (**1**) by using identical conditions of hydrogel (**5**) preparation. The formation of hydrogels and all gel precursors are confirmed by FT-IR and NMR spectroscopy. The theoretical studies of hydrogels and all precursor compounds offer evidence for their successful synthesis, reactivities and potential biological activities. The concept of the present

work could potentially be used to fabricate well-defined hydrogel from various polymeric systems to overcome many persisting drawbacks of the conventional hydrogels, including fast stimuli sensitivity, mechanical strength and so forth. The hydrogel fabricated using click reaction can be post-treated to have desired functionalities, which may open up a new horizon in biomedical fields. The concept can also be used to join polymer segments having the same and different functionality to obtain smart polymers.

Data accessibility. All the datasets supporting this article are available at the Dryad Digital Repository: https://doi.org/10.5061/dryad.5x69p8d1v [40].

Authors' contributions. A.B.I. designed and supervised the project, and wrote the manuscript. M.K.A. carried out all experiments and wrote the draft manuscript. A.K. performed computational analysis and wrote the draft manuscript.

Competing interests. We declare we have no competing interests

Acknowledgements. A.B.I. gratefully acknowledges the support of the Grant of Advanced Research in Education (GARE) (PS2016239) from the Ministry of Education, the Peoples Republic of Bangladesh. The author is also thankful to Committee for Advanced Studies and Research (CASR) in BUET for funding.

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
