## [Peer Review File · Royal Society Open Science]

Review History

RSOS-202056.R0 (Original submission)

Review form: Reviewer 1

Is the manuscript scientifically sound in its present form?

Yes

Are the interpretations and conclusions justified by the results?

Yes

Is the language acceptable?

No

Do you have any ethical concerns with this paper?

No

Have you any concerns about statistical analyses in this paper?

No

Recommendation?

Major revision is needed (please make suggestions in comments)

Comments to the Author(s)

Comments to the Author

In this manuscript, the authors prepared hydrogel using simple click reaction between azide derivative polymer and telechelic acetylene precursor with a copper catalyst. The optimized structure of each precursor is determined, and their chemical and thermodynamic parameters are computationally studied in detail. The concept of the present work could potentially be used to fabricate well defined hydrogel from various polymeric systems to overcome many persisting drawbacks of the conventional hydrogels. If the concept can be used to join polymer segments o obtain sustainable polymers, it is will be very interesting in nowadays. My detailed comments are as follows:

- 1、 First of all, I think the title is inappropriate. Step-wise synthesis is a series of reactions, not a quick success of click-chemistry, although this reaction is the most critical
- 2、 Page 4, L8-16. "Physical networks" and "chemical crosslinking" would be better to be confirmed and explained using some literature.
- 3、 Page 7, L34-36. Synthetic hydrogel (5) , detection and characterization were too few, only infrared, more characterization and detection would better demonstrate the polymer formed.
- 4、 In Figure 1.b. Corresponding peaks and functional groups should be corresponded and marked on the Figure.
- 5、 Figure 2.a. Compound 3 and 4 should have infrared absorption characteristic peaks of azide groups, not indicated in the figure or below.
- 6、 Page 13, L42-45. Compound 4 no absorption band observed for C=C bond in the FT-IR spectrum, are not consistent obviously compared to other compounds.
- 7、 Figure 6. Swelling ratios and thermogravimetric analysis of hydrogelc (5)/(9) is a little bit different. However, it lacks corresponding analysis and theoretical basis.
8. If the obtained polymer was applied in some case with the better properties compared to other polymer, it would be increase the capacity of the polymer functioned as a novel materials.

Review form: Reviewer 2

Is the manuscript scientifically sound in its present form?

Yes

Are the interpretations and conclusions justified by the results?

Yes

Is the language acceptable?

Yes

Do you have any ethical concerns with this paper?

No

Have you any concerns about statistical analyses in this paper?

No

Recommendation?

Accept with minor revision (please list in comments)

Comments to the Author(s)

Manuscript ID: RSOS-202056

Title: Step-wise synthesis of polymer network using click chemistry and their computational study

Recommendation: Minor Revision

Comments:

In this manuscript, polymer networks have been synthesized via two different routes and its computational study was performed. Lots of synthesis works done and all intermediates, polymers, and final product, hydrogel are perfectly characterized by FTIR and proton NMR techniques. The present work is meaningful, interesting, and the manuscript is also well organized. Figures are well presented. It can be published after some revisions.

1. Abstract is the most important part of an article. It should be precise and easy to understand. Here it seems a bit lengthy and the first sentence itself is too long. Moreover, DCC/DMAP and AIBN should explain as it comes for the first time in the manuscript. In sentences regarding route 1 and 2, please correct sentence structures such as using the DCC/DMAP catalyst or using DCC/DMAP as a catalyst, replacing bromide by azide moiety, modification of the polymer with azide group via successive steps to obtain azide derivative polymer for click reaction, using copper as a catalyst, etc. In "The derivatives of azide functionalized HEMA, acetylene precursors, and hydrogels are confirmed by FTIR and ¹H-NMR spectroscopy." What is confirmed? Structures?

Reply:

2. "Computational Chemistry" is very broad, so it cannot be a keyword. Please replace it with a suitable one.

Reply:

3. Introduction is properly written with the perfect length but should include few more recent references.

Reply:

4. In the synthesis section is it necessary to mention the FTIR bands and proton NMRs corresponding chemical shift values? As is mentioned in the results and discussion section.

Reply:

5. In the synthesis of compound 2, "The insoluble compound was separated by filtration, and the filtrate was extracted with dichloromethane followed by washing multiple times with saturated NaHCO₃ aqueous solution and pH 1 aqueous solution in a separatory funnel". Meaning of this sentence is not clear. What is round bottom?

Reply:

6. Page 12, line 12: "The dichloromethane was found not a suitable solvent for this reaction, as at high temperatures, the azide coupling reactions stopped there." Explain this sentence.

Reply:

7. Page 14, line 42: "The small energy difference of HOMO-LUMO (ΔE) is considered as high chemical stability of compounds, which leads to their biomedical applications." Please add suitable references.

Reply:

8. Why authors employed two different routes for the polymer synthesis? Explain with comparative data.

Reply:

9. Table S1 and S2 units should be in the column heading. Table S4 mentions the unit.

Reply:

Decision letter (RSOS-202056.R0)

Dear Dr Imran:

Title: Step-wise synthesis of polymer network using click chemistry and their computational study

Manuscript ID: RSOS-202056

The editor assigned to your manuscript has now received comments from reviewers. We would like you to revise your paper in accordance with the referee and Subject Editor suggestions which can be found below (not including confidential reports to the Editor). Please note this decision does not guarantee eventual acceptance.

Please submit your revised paper before 02-Jan-2021. Please note that the revision deadline will expire at 00.00am on this date. If we do not hear from you within this time then it will be assumed that the paper has been withdrawn. In exceptional circumstances, extensions may be possible if agreed with the Editorial Office in advance. We do not allow multiple rounds of revision so we urge you to make every effort to fully address all of the comments at this stage. If deemed necessary by the Editors, your manuscript will be sent back to one or more of the original reviewers for assessment. If the original reviewers are not available we may invite new reviewers.

On behalf of the Subject Editor Professor Anthony Stace and the Associate Editor Professor Chaohua Cui.

RSC Associate Editor:
Comments to the Author:
(There are no comments.)

RSC Subject Editor:
Comments to the Author:
(There are no comments.)

Reviewers' Comments to Author:
Reviewer: 1

Comments to the Author(s)
Comments to the Author

In this manuscript, the authors prepared hydrogel using simple click reaction between azide derivative polymer and telechelic acetylene precursor with a copper catalyst. The optimized structure of each precursor is determined, and their chemical and thermodynamic parameters are computationally studied in detail. The concept of the present work could potentially be used to fabricate well defined hydrogel from various polymeric systems to overcome many persisting drawbacks of the conventional hydrogels. If the concept can be used to join polymer segments or obtain sustainable polymers, it will be very interesting in nowadays. My detailed comments are as follows:

- 1、 First of all, I think the title is inappropriate. Step-wise synthesis is a series of reactions, not a quick success of click-chemistry, although this reaction is the most critical
- 2、 Page 4, L8-16. "Physical networks" and "chemical crosslinking" would be better to be confirmed and explained using some literature.
- 3、 Page 7, L34-36. Synthetic hydrogel (5), detection and characterization were too few, only infrared, more characterization and detection would better demonstrate the polymer formed.
- 4、 In Figure 1.b. Corresponding peaks and functional groups should be corresponded and marked on the Figure.
- 5、 Figure 2.a. Compound 3 and 4 should have infrared absorption characteristic peaks of azide groups, not indicated in the figure or below.
- 6、 Page 13, L42-45. Compound 4 no absorption band observed for C=C bond in the FT-IR spectrum, are not consistent obviously compared to other compounds.
- 7、 Figure 6. Swelling ratios and thermogravimetric analysis of hydrogelc (5)/(9) is a little bit different. However, it lacks corresponding analysis and theoretical basis.
8. If the obtained polymer was applied in some case with the better properties compared to other polymer, it would be increase the capacity of the polymer functioned as a novel materials.

Reviewer: 2

Comments to the Author(s)
Manuscript ID: RSOS-202056

Title: Step-wise synthesis of polymer network using click chemistry and their computational study

Recommendation: Minor Revision

Comments:

In this manuscript, polymer networks have been synthesized via two different routes and its computational study was performed. Lots of synthesis works done and all intermediates, polymers, and final product, hydrogel are perfectly characterized by FTIR and proton NMR techniques. The present work is meaningful, interesting, and the manuscript is also well organized. Figures are well presented. It can be published after some revisions.

1. Abstract is the most important part of an article. It should be precise and easy to understand. Here it seems a bit lengthy and the first sentence itself is too long. Moreover, DCC/DMAP and AIBN should explain as it comes for the first time in the manuscript. In sentences regarding route 1 and 2, please correct sentence structures such as using the DCC/DMAP catalyst or using DCC/DMAP as a catalyst, replacing bromide by azide moiety, modification of the polymer with azide group via successive steps to obtain azide derivative polymer for click reaction, using copper as a catalyst, etc. In "The derivatives of azide functionalized HEMA, acetylene precursors, and hydrogels are confirmed by FTIR and ¹H-NMR spectroscopy." What is confirmed? Structures?

Reply:

2. "Computational Chemistry" is very broad, so it cannot be a keyword. Please replace it with a suitable one.

Reply:

3. Introduction is properly written with the perfect length but should include few more recent references.

Reply:

4. In the synthesis section is it necessary to mention the FTIR bands and proton NMRs corresponding chemical shift values? As is mentioned in the results and discussion section.

Reply:

5. In the synthesis of compound 2, "The insoluble compound was separated by filtration, and the filtrate was extracted with dichloromethane followed by washing multiple times with saturated NaHCO₃ aqueous solution and pH 1 aqueous solution in a separatory funnel". Meaning of this sentence is not clear. What is round bottom?

Reply:

6. Page 12, line 12: "The dichloromethane was found not a suitable solvent for this reaction, as at high temperatures, the azide coupling reactions stopped there." Explain this sentence.

Reply:

7. Page 14, line 42: "The small energy difference of HOMO-LUMO (ΔE) is considered as high chemical stability of compounds, which leads to their biomedical applications." Please add suitable references.

Reply:

8. Why authors employed two different routes for the polymer synthesis? Explain with comparative data.

Reply:

9. Table S1 and S2 units should be in the column heading. Table S4 mentions the unit.

Reply:

Author's Response to Decision Letter for (RSOS-202056.R0)

See Appendix A.

RSOS-202056.R1 (Revision)

Review form: Reviewer 1

Is the manuscript scientifically sound in its present form?

Yes

Are the interpretations and conclusions justified by the results?

No

Is the language acceptable?

No

Do you have any ethical concerns with this paper?

No

Have you any concerns about statistical analyses in this paper?

No

Recommendation?

Accept with minor revision (please list in comments)

Comments to the Author(s)

Some characterization such SEM should be used to characterize the morphology of hydrogel.

Review form: Reviewer 2

Is the manuscript scientifically sound in its present form?

Yes

Are the interpretations and conclusions justified by the results?

Yes

Is the language acceptable?

Yes

Do you have any ethical concerns with this paper?

No

Have you any concerns about statistical analyses in this paper?

No

Recommendation?

Accept as is

Comments to the Author(s)

Thank you. You have well addressed all my concerns.

Decision letter (RSOS-202056.R1)

Dear Dr Imran:

Title: Facile fabrication of polymer network using click chemistry and their computational study
Manuscript ID: RSOS-202056.R1

Thank you for submitting the above manuscript to Royal Society Open Science. On behalf of the Editors and the Royal Society of Chemistry, I am pleased to inform you that your manuscript will be accepted for publication in Royal Society Open Science subject to minor revision in accordance with the referee suggestions. Please find the reviewers' comments at the end of this email.

The reviewers and handling editors have recommended publication, but also suggest some minor revisions to your manuscript. Therefore, I invite you to respond to the comments and revise your manuscript.

Because the schedule for publication is very tight, it is a condition of publication that you submit the revised version of your manuscript before 05-Feb-2021. Please note that the revision deadline will expire at 00.00am on this date. If you do not think you will be able to meet this date please let me know immediately.

- 1) A text file of the manuscript (tex, txt, rtf, docx or doc), references, tables (including captions) and figure captions. Do not upload a PDF as your "Main Document".
- 2) A separate electronic file of each figure (EPS or print-quality PDF preferred (either format should be produced directly from original creation package), or original software format)
- 3) Included a 100 word media summary of your paper when requested at submission. Please ensure you have entered correct contact details (email, institution and telephone) in your user account
- 4) Included the raw data to support the claims made in your paper. You can either include your data as electronic supplementary material or upload to a repository and include the relevant doi within your manuscript
- 5) All supplementary materials accompanying an accepted article will be treated as in their final form. Note that the Royal Society will neither edit nor typeset supplementary material and it will

be hosted as provided. Please ensure that the supplementary material includes the paper details where possible (authors, article title, journal name).

Kind regards,

Dr Laura Smith
Publishing Editor, Journals

On behalf of the Subject Editor Professor Anthony Stace and the Associate Editor Professor Chaohua Cui.

RSC Associate Editor:
Comments to the Author:
(There are no comments.)

RSC Subject Editor:
Comments to the Author:
(There are no comments.)

Reviewer comments to Author:
Reviewer: 1

Comments to the Author(s)
Some characterization such SEM should be used to characterize the morphology of hydrogel.

Reviewer: 2

Comments to the Author(s)
Thank you. You have well addressed all my concerns.

Author's Response to Decision Letter for (RSOS-202056.R1)

See Appendix B.

Decision letter (RSOS-202056.R2)

Dear Dr Imran:

Title: Facile fabrication of polymer network using click chemistry and their computational study
Manuscript ID: RSOS-202056.R2

It is a pleasure to accept your manuscript in its current form for publication in Royal Society Open Science. The chemistry content of Royal Society Open Science is published in collaboration with the Royal Society of Chemistry.

On behalf of the Subject Editor Professor Anthony Stace and the Associate Editor Professor Chaohua Cui.

RSC Associate Editor
Comments to the Author:
(There are no comments.)

Reviewer(s)' Comments to Author:

Appendix A

Reviewers' Comments to Author:

Reviewer: 1

Comments to the Author(s)

In this manuscript, the authors prepared hydrogel using simple click reaction between azide derivative polymer and telechelic acetylene precursor with a copper catalyst. The optimized structure of each precursor is determined, and their chemical and thermodynamic parameters are computationally studied in detail. The concept of the present work could potentially be used to fabricate well defined hydrogel from various polymeric systems to overcome many persisting¹⁻⁵. If the concept can be used to join polymer segments or obtain sustainable polymers, it will be very interesting in nowadays.

Response: Thank you for your careful and prudent reviewing of the manuscript. We are very thankful for your valuable comments and suggestions. Please find replies to your comments and an outline of revision.

My detailed comments are as follows:

1、 First of all, I think the title is inappropriate. Step-wise synthesis is a series of reactions, not a quick success of click-chemistry, although this reaction is the most critical

Reply: Cordial thanks for your suggestions. The title of the manuscript has been changed according to your recommendation as- Facile fabrication of polymer network using click chemistry and their computational study.

2、 Page 4, L8-16. “Physical networks” and “chemical crosslinking” would be better to be confirmed and explained using some literature.

Reply: Yes, you are right. The following references are cited in the revised manuscript

a) Design, development, characterization and application of smart polymeric hydrogel, Abu Bin Imran in *Manufacturing Systems: Recent Progress and Future Directions*, Dr. Mohamed Arezki Mellal (Editor), *Nova Science Publishers, Inc.*, 400 Oser Avenue, Suite 1600, Hauppauge, NY 11788-3619, USA, 2020 (Accepted book chapter)

b) S. R. V. Tomme, M. J.V. Steenbergen, S. C. D. Smedt, C. F. V. Nostrum, W. E. Hennink, *Biomaterials*, 2005, **26**, 2129–2135.

c) C. L. Bell, N. A., Peppas, *Journal of Controlled Release*, 1996, **39**, 201–207.

d) W. S. Dai, T. A. Barbari, *Journal of Membrane Science*, 1999, **156**, 67–79.

e) E., Jabbari, S. Nozari, *European Polymer Journal*, 2000, **36**, 2685–2692.

f) R. Kishi, O. Hirasa, H. Ichijo, *Polymer Gels and Networks*, 1997, **5**, 145–151.

3、 Page 7, L34-36. Synthetic hydrogel (5), detection and characterization were too few, only infrared, more characterization and detection would better demonstrate the polymer formed.

Reply: Thank you so much. The physical appearance and the amount of consumed gel precursors are the testament of hydrogel formation. In our case the hydrogel was formed after click reaction by consuming almost 100% of gel precursors. In addition, the swelling behavior of hydrogel (5) has been described in the manuscript. The insoluble nature of hydrogel (5) in various solvents are also discussed. All of these results clearly indicate the successful hydrogel formation.

4、 In Figure 1.b. Corresponding peaks and functional groups should be corresponded and marked on the Figure.

Reply: Yes, without any iota of doubt, we agree with your comment. The peak at 1687 cm^{-1} for C=O, the peak at 2945 cm^{-1} for C-H, the peak at 2131 cm^{-1} for $\text{C}\equiv\text{C}$, the peak at 3310 cm^{-1} for N-H have been marked and discussed in the revised manuscript.

5、 Figure 2.a. Compound 3 and 4 should have infrared absorption characteristic peaks of azide groups, not indicated in the figure or below.

Reply: Infrared absorption of azide group for Compound 3 is at 2120 cm^{-1} , for Compound 4 at 2124 cm^{-1} and for compound 8 at 2118 cm^{-1} , respectively are observed after magnification of the curves in that region. Please check below. Due to low intensity of this peak, it is not clearly visible in overlay figure. It has been mentioned in the revised manuscript. Thanks for your understanding.

Magnified FTIR curve of compound 8

6、 Page 13, L42-45. Compound 4 no absorption band observed for C=C bond in the FT-IR spectrum are not consistent obviously compared to other compounds.

Reply: Thank so much for the very interesting concern. Actually, disappearance of absorption band of C=C bond of the compound **4** in the FT-IR spectrum prove the successful polymerization of monomers. If we compare with compounds (monomers) like **2**, **3**, the IR spectra of monomers give C=C bond peaks.

7、 Figure 6. Swelling ratios and thermogravimetric analysis of hydrogelc (5)/(9) is a little bit different. However, it lacks corresponding analysis and theoretical basis.

Reply: Thanks again. Thermogravimetric analysis of hydrogels shows that both hydrogels exhibit a notable thermal stability for the presence of triazole ring (coupling by azides and alkynes) as central core. Long chain length in polymer moiety also increases the thermal stability of hydrogels.

8. If the obtained polymer was applied in some case with the better properties compared to other polymer, it would be increase the capacity of the polymer functioned as a novel materials.

Reply: Yes, you are right. The concept can be applied to many polymeric systems specially in hydrogels and elastomers. Our group is working hard to materialize their manifold applications. Hopefully, we would be able to communicate further soon with some more interesting findings.

Reviewer: 2

Comments to the Author(s)

Manuscript ID: RSOS-202056

Title: Step-wise synthesis of polymer network using click chemistry and their computational study

Recommendation: Minor Revision

Comments:

In this manuscript, polymer networks have been synthesized via two different routes and its computational study was performed. Lots of synthesis works done and all intermediates, polymers, and final product, hydrogel are perfectly characterized by FTIR and proton NMR techniques. The present work is meaningful, interesting, and the manuscript is also well organized. Figures are well presented. It can be published after some revisions.

Response: Thank you for your careful and prudent reviewing of the manuscript. We are very thankful for your valuable comments and suggestions. Please find replies to your comments and an outline of revision.

1. Abstract is the most important part of an article. It should be precise and easy to understand. Here it seems a bit lengthy and the first sentence itself is too long.

Reply: Thanks a lot for pointing out the interesting corrections in the abstract section.

The first sentence has been rephrased in the revised manuscript.

Click reaction is a very fast, high yield with no byproduct, biocompatible, tolerant to surrounded medium, and very specific cycloaddition reaction between azides and alkynes to form triazole. They are and widely being employed in the synthesis of various polymeric materials.

Moreover, DCC/DMAP and AIBN should explain as it comes for the first time in the manuscript.

The details of DCC: N, N'-Dicyclohexylcarbodiimide; DMAP: 4-Dimethylaminopyridine; and AIBN: Azobisisobutyronitrile have been incorporated in the revised manuscript.

In sentences regarding route 1 and 2, please correct sentence structures such as using the DCC/DMAP catalyst or using DCC/DMAP as a catalyst, replacing bromide by azide moiety, modification of the polymer with azide group via successive steps to obtain azide derivative polymer for click reaction, using copper as a catalyst, etc.

All corrections have been made in the revised manuscript.

In “The derivatives of azide functionalized HEMA, acetylene precursors, and hydrogels are confirmed by FTIR and ¹H-NMR spectroscopy.” What is confirmed? Structures?

Thanks a lot for pointing out the unintentional mistake. Yes we confirmed the structures by FTIR and ¹H-NMR spectroscopy. In the revised manuscript we mentioned that

The structures of derivatives of azide functionalized HEMA, acetylene precursors, and hydrogels are confirmed by FTIR and ¹H-NMR spectroscopy.

2. “Computational Chemistry” is very broad, so it cannot be a keyword. Please replace it with a suitable one.

Reply: We replace the keywords with HOMO, and LUMO.

3. Introduction is properly written with the perfect length but should include few more recent references.

Reply: Thanks again. The following recent references have been added in the reference section
a) M. Meldal, F. Diness, “Recent Fascinating Aspects of the CuAAC Click Reaction”, *Trends in Chemistry*, 2020, 2(6), 569-584.

b) Neumann, S., Biewend, M., Rana, S., Binder, W. H., The CuAAC: Principles, Homogeneous and Heterogeneous Catalysts, and Novel Developments and Applications. *Macromol. Rapid Commun.* 2020, 41, 1900359. <https://doi.org/10.1002/marc.201900359>

c) Yury Golitsyn, Martin Pulst, Muhammad Haris Samiullah, Karsten Busse, Jörg Kressler, Detlef Reichert, Crystallization in PEG networks: The importance of network topology and chain tilt in crystals, *Polymer*, Volume 165, 2019, Pages 72-82, ISSN 0032-3861, <https://doi.org/10.1016/j.polymer.2019.01.018>

4. In the synthesis section is it necessary to mention the FTIR bands and proton NMRs corresponding chemical shift values? As is mentioned in the results and discussion section.

Reply: Thank you. In the experimental section, the details of IR, NMR are appended but, in the results, and discussion only the highlighted peaks are mentioned.

5. In the synthesis of compound 2, “The insoluble compound was separated by filtration, and the filtrate was extracted with dichloromethane followed by washing multiple times with saturated NaHCO₃ aqueous solution and pH 1 aqueous solution in a separatory funnel”. Meaning of this sentence is not clear.

Reply: Thank you. Here we employed solvent extraction technique to purify the product.

What is round bottom?

We are sorry for this unintentional mistake. It will be “round bottom flask”. Round-bottom flasks are used in distillation by chemists as distilling flasks and receiving flasks for the distillate (see distillation diagram).

6. Page 12, line 12: “The dichloromethane was found not a suitable solvent for this reaction, as at high temperatures, the azide coupling reactions stopped there.” Explain this sentence.

Reply: Thank you again. The insertion of azide into compound **2** was not successful when we used dichloromethane as solvent at high temperature. The click reaction is highly dependent on the reaction medium.

7. Page 14, line 42: “The small energy difference of HOMO-LUMO (ΔE) is considered as high chemical stability of compounds, which leads to their biomedical applications.” Please add suitable references.

Reply: Cordial thanks for your suggestion. The following references are added in the revised manuscript.

1. Mihçioğur, Ö.Ö., Talat, Molecular structure, vibrational spectroscopic analysis (IR & Raman), HOMO-LUMO and NBO analysis of anti-cancer drug sunitinib using DFT method. *Journal of Molecular Structure*, 2017. 1149: p. 27-41.

2. Xavier, T., N. Rashid, and I.H. Joe, Vibrational spectra and DFT study of anticancer active molecule 2-(4-Bromophenyl)-1H-benzimidazole by normal coordinate analysis. *Spectrochimica Acta Part A: Molecular and Biomolecular Spectroscopy*, 2011. 78(1): p. 319-326.

3. Aihara, J.-i., Reduced HOMO– LUMO gap as an index of kinetic stability for polycyclic aromatic hydrocarbons. *The Journal of Physical Chemistry A*, 1999. 103(37): p. 7487-7495.

8. Why authors employed two different routes for the polymer synthesis? Explain with comparative data.

Reply: To explore the material properties of hydrogel from two approach we choose two different routes for the polymer synthesis though. But no significant variation of material properties is observed for two hydrogels. Either method successfully produce polymer networks.

9. Table S1 and S2 units should be in the column heading. Table S4 mentions the unit.

Reply: Thank you. The units of Table S1 and S2 units have been replaced in the column heading and unit for Table S4 has been added.

Appendix B

Reviewer comments to Author:

Reviewer: 1

Comments to the Author(s)

Some characterization such SEM should be used to characterize the morphology of hydrogel.

Response: Thank you for your careful and prudent reviewing of the manuscript. We are very thankful for your valuable comments and suggestions which helped us a lot to improve the quality of the article. Please find replies to your comments regarding morphology characterization of hydrogels using SEM.

As you know that, the hydrogel reported in this study was prepared solely by crosslinking between telechelic functional compounds and monomer/polymer by click reaction. We have long experience on morphology characterization of such hydrogel system. Unfortunately, similar types of hydrogels are unable to produce any significant/important morphology information from SEM unlike nanocomposite hydrogel. Even after tremendous treatment and precautions, microlevel information can only be achieved from SEM. As I guess you are rightly expecting some interesting information from their point of view of formation of homogeneous network. We are completely agreeing on that point. We have a strong intention to further characterize in details of our hydrogels using Small angle Neutron Scattering (SANS) and Small angle X-ray Scattering (SAXS). For example:

Chang LIU, Hiroaki Gotoh, Imran Abu Bin, Mitsuo Hara, Takahiro Seki, Koichi Mayumi, Kohzo Ito, Yukikazu Takeoka. Optically Transparent, High Toughness Elastomer Using a Polyrotaxane Cross-linker as a Molecular Pulley, Science Advances, 4 (10), 1-9 (2018). DOI: 10.1126/sciadv.aat7629

Imran A. B, Esaki, K., Gotoh, H., Seki, T., Ito, K., Sakai, Y., Takeoka, Y. "Extremely stretchable thermosensitive hydrogels by introducing slide-ring polyrotaxane cross-linkers and ionic groups into the polymer network" Nature Communications, 5, 5124, 1-8 (2014).

Hopefully we will soon communicate this probable interesting information to publish in another manuscript. Your comment really opens up our eyes to work a new direction on this project. Thanks again.

Reviewer: 2

Comments to the Author(s)

Thank you. You have well addressed all my concerns.

Response: Thank you for your careful and prudent reviewing of the manuscript. We are very thankful for your valuable comments and suggestions which helped us a lot to improve the quality of the article.